

# Genetic diversity of Simao pine in China revealed by SRAP markers

Dawei Wang[1,2], Bingqi Shen[1,2] and Hede Gong[3]

[1] Key Laboratory for Forest Resource Conservation and Utilization in the Southwest Mountains of China, Ministry of Education, Southwest Forestry University, Kunming, Yunnan, China
[2] Key Laboratory for Forest Genetic and Tree Improvement & Propagation in Universities of Yunnan Province, Southwest Forestry University, Kunming, Yunnan, China
[3] School of Geography, Southwest Forestry University, Kunming, Yunnan, China

## ABSTRACT

**Background**. Simao pine (*Pinus kesiya* Royle ex Gordon var. *langbianensis* (A. Chev.) Gaussen) is one of the most important tree species in the production of timber and resin in China. However, the genetic diversity of the natural populations has not been assessed to date. In this study, sequence related amplified polymorphism (SRAP) markers were used to investigate the genetic composition of natural Simao pine populations.

**Method**. The SRAP markers were applied and their efficiency was compared using various statistical multivariate methods, including analysis molecular of variance (AMOVA), the unweighted pair group method with arithmetic mean (UPGMA), and Principal coordinate analysis (PCoA).

**Results**. The 11 populations revealed a high level of genetic diversity (PPB = 95.45%, H = 0.4567, I = 0.6484) at the species level. A moderately low level of genetic differentiation ($G_{st}$ = 0.1701), and a slightly high level of gene flow ($N_m$ = 2.4403) were observed among populations using AMOVA. Eleven populations of Simao pine were gathered into four distinct clusters based on molecular data, and the results of UPGMA and PCoA also illustrated that assignment of populations is not completely consistent with geographic origin. The Mantel test revealed there was no significant correlation between geographic and genetic distance ($r$ = 0.241, $p$ = 0.090).

**Discussion**. The SRAP markers were very effective in the assessment of genetic diversity in Simao pine. Simao pine populations display high levels of genetic diversity and low or moderate levels of genetic differentiation due to frequent gene exchange among populations. The low genetic differentiation among populations implied that conservation efforts should aim to preserve all remaining natural populations of this species. The information derived from this study is useful when identifying populations and categorizing their population origins, making possible the design of long term management program such as genetic improvement by selective breeding.

Corresponding author
Hede Gong, gonghede3@163.com

# INTRODUCTION

Simao pine (*Pinus kesiya* Royle ex Gordon var. *langbianensis* (A. Chev.) Gaussen), a geographic variant of *Pinus kesiya*, is distributed naturally in the humid and sub-humid

areas of the tropical and subtropical zone within Yunnan Province (*Wang et al., 2012a*; *Wang et al., 2012b*). The species is one of the most important timber and resin production trees in China (*Chen, Zhao & Wang, 2002*; *Cai et al., 2017*; *Zhu et al., 2017*; *Wang et al., 2018*), and also has ecological and social benefits, in addition to economic ones (*Ou et al., 2016*; *Li et al., 2018a*; *Li et al., 2018b*). There is high potential for over utilization of the tree species and Simao pine breeding programs are constrained by a narrow genetic background and overexploitation (*Zhao et al., 2016*). Genetic improvement will be salient to future Simao pine breeding and in order to selectively breed Simao pine in a coordinated breeding scheme, genetic diversity analysis needs to be implemented.

Sequence-related amplified polymorphism (SRAP) is a kind of molecular marker technology based on polymerase chain reaction (PCR). The method is convenient, exhibits high co-dominance, is uncomplicated in the separation of strips and sequencing, and doesn't require knowledge of the sequence information of species in advance (*Li & Quiros, 2001*). The SRAP markers are simple, reliable, easily detected, genome specific, highly polymorphic and commonly used in genomic applications (*Ma et al., 2015*). This form of analysis is also very effective for the assessment of genetic diversity (*Shaye et al., 2018*). For these reasons, SRAP markers are widely used in population genetic analysis of various plant species (*Liu et al., 2016*; *Bhatt et al., 2017*; *Li et al., 2018a*; *Li et al., 2018b*).

In this study, SRAP markers were used to investigate the genetic composition of natural Simao pine populations, which are limitedly distributed in Southwest Yunnan, China. The study's aims were to evaluate genetic diversity at population and species levels in the Simao pine; assess the distribution of the genetic variation within and among populations, and construct a dendrogram demonstrating the genetic relationships among them. The genetic information can then be used as a tool for assessing the current conservation management plan for this species and for designing conservation strategies.

## MATERIAL AND METHODS

### Plant materials

Eleven natural populations of Simao pine with a total of 290 individuals were sampled throughout the species distribution range (Fig. 1 and Table 1). Field experiments were approved by Southwest Forestry University (project number: 2013Y121). In large populations ($n > 100$), 30 adult individuals were selected within a distance of $> 100$ m. In small populations ($n < 30$), all available adult individuals were sampled. For each sampled individual, young fresh pine needles ($\sim 10$ g) were collected, dried in silica gel, and stored at $-80\,°C$ until subsequent DNA extraction.

### DNA extraction

Genomic DNA was isolated from the pine needles of each individual tree according to the cetyltrimethyl ammonium bromide (CTAB) method (*Wang, Li & Li, 2011*). All genomic DNA was kept frozen at $-20\,°C$ for standby application.

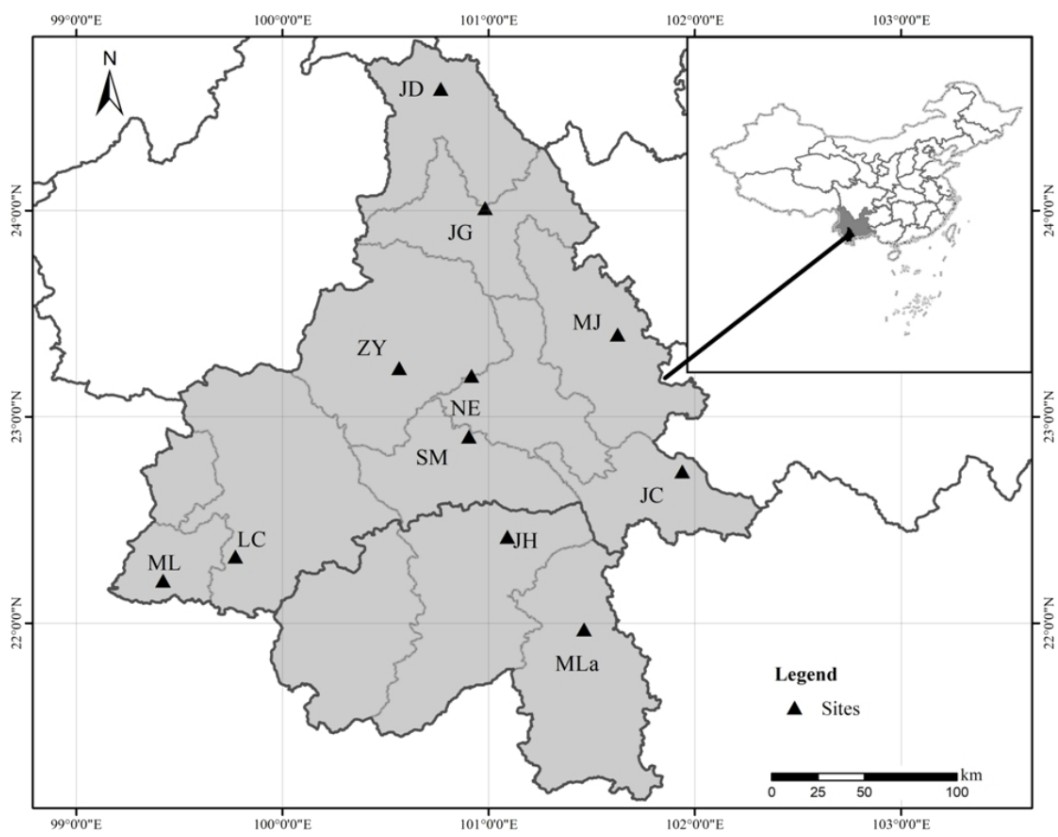

**Figure 1** Geographical location of the 11 Natural populations of Simao pine in China.

**Table 1** Locations and geographic characteristics of the sampled Simao pine populations.

| Population name | Population abbreviation | Longitude (°E) | Latitude (°N) | Altitude (m) | $T_{mean}$ (°C) | $T_{max}$ (°C) | $T_{min}$ (°C) | Pr (mm) | $Pr_{max}$ (mm) | $Pr_{min}$ (mm) | Number |
|---|---|---|---|---|---|---|---|---|---|---|---|
| Mojiang | MJ | 101.62 | 23.40 | 1,578 | 19.0 | 28.2 | 5.9 | 279 | 1,322 | 15 | 30 |
| Ninger | NE | 100.91 | 23.19 | 1,383 | 20.1 | 29.7 | 6.4 | 283 | 1,342 | 13 | 30 |
| Jingdong | JD | 100.76 | 24.59 | 1,485 | 19.0 | 28.5 | 5.3 | 191 | 965 | 13 | 30 |
| Jinggu | JG | 100.98 | 24.01 | 1,525 | 19.8 | 29.5 | 5.8 | 285 | 1,342 | 13 | 30 |
| Zhenyuan | ZY | 100.58 | 23.23 | 1,964 | 16.8 | 25.9 | 3.3 | 238 | 1,144 | 14 | 30 |
| Jiangcheng | JC | 101.93 | 22.73 | 1,263 | 16.9 | 25.6 | 4.4 | 398 | 1,772 | 22 | 30 |
| Lancang | LC | 99.77 | 22.32 | 1,725 | 17.8 | 27.4 | 3.6 | 314 | 1,520 | 12 | 30 |
| Menglian | ML | 99.42 | 22.21 | 1,239 | 18.7 | 28.5 | 4.4 | 283 | 1,449 | 9 | 30 |
| Jinghong | JH | 101.09 | 22.41 | 870 | 20.6 | 30.2 | 7.6 | 290 | 1,434 | 15 | 20 |
| Mengla | Mla | 101.46 | 21.96 | 1,032 | 19.0 | 28.0 | 6.4 | 319 | 1,645 | 20 | 20 |
| Simao | SM | 100.90 | 22.90 | 1,355 | 18.4 | 27.9 | 4.3 | 324 | 1,491 | 14 | 10 |

**Table 2 Primer sequences used for SRAP analysis.**

| Primer combination | Sequences (3′–5′) | |
| --- | --- | --- |
| Me1-Em4 | TGAGTCCAAACCGGATA | GACTGCGTACGAATTTGA |
| Me2-Em9 | TGAGTCCAAACCGGAGC | GACTGCGTACGAATTGAG |
| Me4-Em9 | TGAGTCCAAACCGGACC | GACTGCGTACGAATTGAG |
| Me5-Em10 | TGAGTCCAAACCGGAAG | GACTGCGTACGAATTGCC |
| Me6-Em2 | TGAGTCCAAACCGGTAA | GACTGCGTACGAATTTGC |
| Me6-Em3 | TGAGTCCAAACCGGTAA | GACTGCGTACGAATTGAC |
| Me6-Em5 | TGAGTCCAAACCGGTAA | GACTGCGTACGAATTAAC |
| Me6-Em9 | TGAGTCCAAACCGGTAA | GACTGCGTACGAATTGAG |
| Me7-Em1 | TGAGTCCAAACCGGTCC | GACTGCGTACGAATTAAT |
| Me7-Em7 | TGAGTCCAAACCGGTCC | GACTGCGTACGAATTCAA |
| Me8-Em4 | TGAGTCCAAACCGGTGC | GACTGCGTACGAATTTGA |
| Me9-Em6 | TGAGTCCAAACCGGAAC | GACTGCGTACGAATTGCA |
| Me9-Em7 | TGAGTCCAAACCGGAAC | GACTGCGTACGAATTCAA |
| Me9-Em9 | TGAGTCCAAACCGGAAC | GACTGCGTACGAATTGAG |
| Me10-Em3 | TGAGTCCAAACCGGTAG | GACTGCGTACGAATTGAC |

## SRAP-PCR Amplification Analysis

Fifteen primer combinations (Table 2), selected from the initial 100 pairs of primer combinations, were used for the study. Next, SRAP-PCRs were run in 25 μL volumes containing 60 ng DNA template, 2.0 mM $MgCl_2$, primer 1 μM, dNTPs 0.2 mM, and 1U *Taq* DNA polymerase (Fermentas, Ottawa, Canada). The following cycling parameters were used for amplification: 4 min denaturing at 94 °C, five cycles of 94 °C for 1 min; 35 °C for 45 s and 72 °C for 1 min, 30 cycles of 94 °C for 1 min, annealing for 1 min, 72 °C for 1 min, and 5 min 72 °C for final extension. The PCR products were stored at −20 °C. Finally, amplification products were separated on 6% denaturing polyacrylamide gel and visualized by silver nitrate staining.

## Data statistics and analysis

The SRAP bands of all 15 pairs of primer combination were graded with presence (1) or absence (0) and transformed into a 1/0 binary character matrix. The binary data matrix was analyzed using the program POPGEN v.1.32 (*Yeh, Yang & Boyle, 1999*) to estimate genetic diversity parameters, including the percentage of polymorphic bands ($P$, %), Nei's gene diversity analysis ($H$), Shannon's information index ($I$), the observed number of alleles ($N_a$), and the effective number of alleles ($N_e$). Total gene diversity ($H_t$), genetic diversity within populations ($H_s$) as well as the relative magnitude of genetic differentiation among populations ($G_{st}$), gene flow ($N_m$), and Nei's genetic distance were also evaluated. Polymorphic information content (PIC) was calculated using a simplified formula (*Anderson et al., 1993*).

Genetic differentiation within and among populations was estimated by the analysis molecular of variance (AMOVA) software package in GenAIEx 6.5 (*Peakall & Smouse, 2006*). To analyze the quality of information from particular SRAP primers, principal component analysis (PCA) was used by GenAIEx 6.5 (*Peakall & Smouse, 2006*). A

dendrogram was generated using the unweighted pair group method with the arithmetic mean (UPGMA) clustering procedure of NTSYS-pc v. 2.02 (*Rohlf, 2000*), and the relationship between geographic and genetic distances was performed with the Mantel test by GenAIEx 6.5 (*Peakall & Smouse, 2006*). We further assessed the genetic structure of populations using the Bayesian clustering approach implemented in the software STRUCTURE v.2.3.3 (*Pritchard, Stephens & Donnelly, 2000*). The number of potential genetic clusters (*K* values) was run from 1 to 20, with 10 independent runs for each *K*. The contribution to the genotypes of the accessions was calculated based on $10^5$ iteration burn-in and $10^5$ iteration sampling periods. Then, the optimal number of clusters *K* was identified following the procedure of *Evanno, Regnaut & Goudet (2005)*.

The ecological data, i.e., annual precipitation (Pr), maximum monthly precipitation ($Pr_{max}$), minimum monthly precipitation ($Pr_{min}$), annual mean temperature ($T_{mean}$), maximum temperature of the warmest month ($T_{max}$), and minimum temperature of the coldest month ($T_{min}$) were extracted from Worldclim-Global Climate Data1 with ArcGIS 9.3 (Table 1, *Hijmans et al., 2005*). Correlations among ecological factors and genetic diversity parameters were determined using a Spearman nonparametric correlation coefficient matrix constructed with SPSS version 18.

# RESULTS

## SRAP-PCR amplification

The selected 15 primer combinations were used to amplify 11 natural populations, and the statistics of amplified primer sites were calculated by the binary character coding method (Table 3). This method was used to amplify 132 sites in the range of 100–1,000 bp. Among these sites were 126 polymorphic loci, the percentage of polymorphism was as high as 95.45%, and each primer amplified an average of 8.8 loci and 8.4 polymorphic loci. This indicated that there was rich genetic diversity among Simao pine, and the genetic background was very complex. The number of polymorphic loci amplified by primer combination Me9-Em6 and Me9-Em7 was 12, which was the highest, and the number of polymorphic loci amplified by primer combination Me4-Em9, Me6-Em3, Me6-Em5, Me9-Em9 was 7, which was the least. In the 15 primer combinations, the percentage of polymorphic loci of primer combination Me6-Em9 and Me8-Em4 were the lowest, at 88%, Me7-Em1 and Me9-Em6 were 90% and 91.66% respectively, and the remaining primer combinations were all 100%. The PIC ranged from 0.43 to 0.77, with an average of 0.63.

## Genetic diversity analysis

At the population level, the percentage of polymorphic bands (*P*) ranged from 77.27% (A) to 100% (C), with an average of 93.46%, and 100.0% at species level. The mean observed number of alleles ($N_a$) ranged from 1.7727 to 2.0000, while the mean effective number of alleles ($N_e$) varied from 1.5469 to 1.8181. Nei's genetic diversity (*H*) varied from 0.3089 to 0.4418, with an average of 0.3801, and Shannon's information indices (*I*) ranged from 0.4511 to 0.6310, with an average of 0.5529. At species level, *H* and *I* were 0.4567 and 0.6484, respectively (Table 4).
**Table 3    Analysis of SRAP-PCR amplification results.**

| Primer combinations | Total number of bands | PB | PPB% | PIC |
|---|---|---|---|---|
| M1-E4 | 9 | 9 | 100 | 0.61 |
| M2-E9 | 9 | 9 | 100 | 0.72 |
| M4-E9 | 7 | 7 | 100 | 0.64 |
| M5-E10 | 10 | 8 | 80 | 0.43 |
| M6-E2 | 8 | 8 | 100 | 0.69 |
| M6-E3 | 7 | 7 | 100 | 0.65 |
| M6-E5 | 7 | 7 | 100 | 0.65 |
| M6-E9 | 9 | 8 | 88.88 | 0.63 |
| M7-E1 | 10 | 9 | 90.00 | 0.51 |
| M7-E7 | 8 | 8 | 100 | 0.56 |
| M8-E4 | 9 | 8 | 88.88 | 0.45 |
| M9-E6 | 12 | 11 | 91.66 | 0.77 |
| M9-E7 | 12 | 12 | 100 | 0.76 |
| M9-E9 | 7 | 7 | 100 | 0.70 |
| M10-E3 | 8 | 8 | 100 | 0.62 |
| Total | 132 | 126 | | |
| Average | 8.8 | 8.4 | 95.45 | 0.63 |

Notes.
(PB), Number of Polymorphic Bands; (PPB%), Number of Percentage of polylnorphic bands; (PIC), Polymorphic Information Content.

**Table 4    Genetic diversity of 11 Simao pine populations.**

| Pop | $N_a$ | $N_e$ | $P$ | $H$ | $I$ | $H_t$ | $H_s$ | $G_{st}$ | $N_m$ |
|---|---|---|---|---|---|---|---|---|---|
| SM | 1.7727 | 1.5469 | 77.27 | 0.3089 | 0.4511 | | | | |
| NE | 1.9394 | 1.6889 | 93.94 | 0.3862 | 0.5611 | | | | |
| MJ | 1.9470 | 1.7274 | 94.70 | 0.4019 | 0.5799 | | | | |
| JD | 1.9394 | 1.6818 | 93.94 | 0.3819 | 0.5555 | | | | |
| JG | 1.9318 | 1.6759 | 93.18 | 0.3781 | 0.5498 | | | | |
| ZY | 1.9091 | 1.6480 | 90.91 | 0.3627 | 0.5288 | | | | |
| JC | 2.0000 | 1.8181 | 100.00 | 0.4418 | 0.6310 | | | | |
| LC | 1.9545 | 1.7072 | 95.45 | 0.3965 | 0.5754 | | | | |
| ML | 1.9545 | 1.6384 | 95.45 | 0.3687 | 0.5430 | | | | |
| JH | 1.9773 | 1.7005 | 97.73 | 0.3945 | 0.5750 | | | | |
| Mla | 1.9545 | 1.6259 | 95.45 | 0.3600 | 0.5313 | | | | |
| Mean | 1.9346 | 1.6781 | 93.46 | 0.3801 | 0.5529 | | | | |
| Total | 2.0000 | 1.8516 | 100.00 | 0.4567 | 0.6484 | 0.4580 | 0.3801 | 0.1701 | 2.4403 |

Notes.
$N_a$, observed number of alleles; $N_e$, effective number of alleles; $P$, percentage of polymorphic loci; $H$, Nei's gene diversity; $I$, Shannon's information indices; $H_t$, total genetic diversity; $H_s$, genetic diversity within populations; $G_{st}$, the relative magnitude of genetic differentiation among populations; $N_m$, gene flow among populations.

**Table 5  Analyses of molecular variance (AMOVA) for Simao pine by SRAP.**

| Source of variation | d.f. | Sum of squares | Variance compent | Percentage of variation% | P value |
|---|---|---|---|---|---|
| Among Pops | 10 | 1,343.086 | 4.146 | 14% | 0.139 |
| Within Pops | 279 | 7,160.500 | 25.665 | 86% | 0.010 |
| Total | 289 | 8,503.586 | 29.810 | 100% | |

**Table 6  Nei's genetic distances and geographical distances among 11 populations.**

| | SM | NE | MJ | JD | JG | ZY | JC | LC | ML | JH | Mla |
|---|---|---|---|---|---|---|---|---|---|---|---|
| SM | **** | 32.76 | 92.15 | 187.44 | 50.61 | 123.16 | 107.94 | 133.22 | 171.17 | 57.27 | 118.53 |
| NE | 0.0826 | **** | 75.92 | 155.03 | 36.23 | 90.48 | 116.85 | 152.635 | 189.02 | 88.41 | 147.32 |
| MJ | 0.1743 | 0.0788 | **** | 158.33 | 109.97 | 94.52 | 80.35 | 224.71 | 262.28 | 121.92 | 159.41 |
| JD | 0.2025 | 0.1214 | 0.0726 | **** | 151.42 | 97.75 | 237.96 | 271.14 | 297.85 | 243.03 | 299.15 |
| JG | 0.2135 | 0.1499 | 0.124 | 0.0661 | **** | 95.92 | 151.44 | 130.17 | 163.98 | 105.53 | 168.09 |
| ZY | 0.2178 | 0.1296 | 0.0861 | 0.0722 | 0.0741 | **** | 172.16 | 224.62 | 256.23 | 177.05 | 231.72 |
| JC | 0.1855 | 0.1135 | 0.0943 | 0.0927 | 0.0948 | 0.082 | **** | 227.86 | 265.85 | 94.26 | 98.16 |
| LC | 0.2529 | 0.1662 | 0.1883 | 0.1969 | 0.1672 | 0.1808 | 0.099 | **** | 38.31 | 136.48 | 178.92 |
| ML | 0.2433 | 0.1538 | 0.1666 | 0.1835 | 0.1694 | 0.1830 | 0.1142 | 0.0649 | **** | 173.71 | 212.45 |
| JH | 0.2511 | 0.1428 | 0.1228 | 0.1605 | 0.1356 | 0.1480 | 0.0879 | 0.0747 | 0.0769 | **** | 62.84 |
| Mla | 0.2820 | 0.1796 | 0.1959 | 0.1993 | 0.1899 | 0.2265 | 0.1299 | 0.1633 | 0.1924 | 0.1433 | **** |

Notes.
Above diagonal: geographical distances (km); below diagonal: genetic distances.

## Genetic differentiation

The total genetic diversity ($H_t$) of the species and genetic diversity within populations ($H_s$) were 0.4580 and 0.3801, respectively (Table 4). The proportion of genetic variation contributed by differences among populations ($G_{st}$) was 0.1701, thus leaving 82.99% of the total genetic variation kept within the populations. The average $N_m$ obtained was 2.4403, suggesting the existence of a certain degree of gene flow among natural distribution populations. This finding was consistent with the results of AMOVA, which detected the highest genetic variation within populations (86%), whereas the variance among populations was only 14% (Table 5).

## Cluster analysis

Based on the SRAP data, a broad range of Nei's genetic distance was found to exist among the 11 Simao pine natural populations, varying from 0.0722 to 0.2820 (Table 6), with a mean of 0.1483. The highest genetic distance pairs were found between population SM and MLa (0.2820), and these populations may be good sources for further breeding purposes. The lowest genetic distance pairs were found between populations JD and ZY (0.0722).

The applied measure of genetic similarity was used to construct UPGMA dendrograms (Fig. 2). The 11 populations were divided into four groups with population SM forming one subgroup; Mla forming another; the populations NE, MJ, JD, JG, ZY and JC forming the third group; and LC, ML and JH were included in the fourth group. The clustering results were not specified completely from location. Furthermore, the dendrogram showed

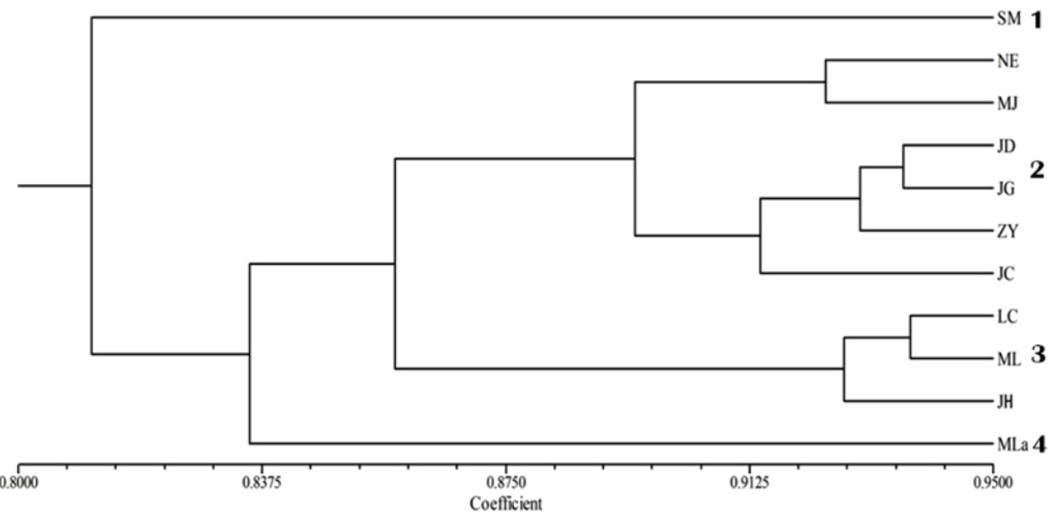

**Figure 2** UPGMA cluster analysis of genetic similarity of 11 populations.

that the populations were partly mixed clusters, although geographically more distant (Fig. 2). Consistent with these results, the Mantel test revealed that there was no significant correlation between geographic and genetic distance ($r = 0.241$, $p = 0.090$).

## Principal coordinate analysis

Principal coordinate analysis (PCoA) was performed to determine the genetic relationships among the accessions with minimum distortion. The first two principal components displayed 37.69% and 60.12% of total variation, respectively, and the 72.74% was expressed by the first three components (Fig. 3). Corresponding with the cluster analysis, each population formed a separate plot and could be clearly distinguished from those of other populations. The first principal coordinate separated 11 populations into two groups. The first group was composed of six high latitude populations including JD, MJ, ZY, JG, NE and SM, and the remaining five southern populations were separated into the second group. Based on the results of the first principal coordinate, the second principal coordinate separated 11 populations into four groups.

## Population structure analysis

The results of the Bayesian clustering analysis of genetic structure showed that the populations analyzed of Simao pine, best fit two genetic groups ($K = 2$, Fig. 4). The percentages of individuals in cluster I and cluster II were 61.87 and 30.38% respectively, whereas 7.75% of the individuals had mixed membership, with $Q$ scores of $0.2 < Q < 0.8$. The individuals in NE, JD, JG, ZY populations with $Q$ values of $>0.8$ belonged to cluster I, and the ML, JH, Mla, SM populations with $Q$ values of $<0.2$ belonged to cluster II. Individuals with admixed population assignments were from MJ, JC and LC populations.

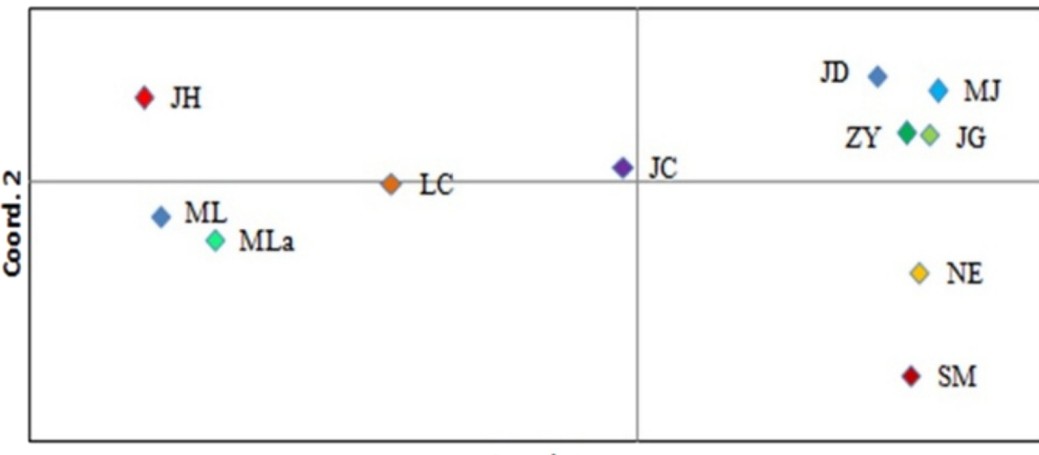

**Figure 3** PCoA of 11 populations based on variability.

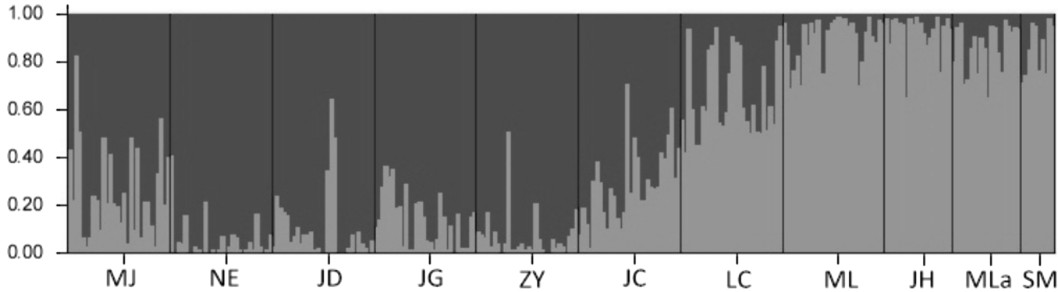

**Figure 4** The proportions of cluster memberships at the individual level in 11 Simao pine populations in the two clusters identified by STRUCTURE.

## Correlation among ecological factors and genetic diversity parameters

Based on the ecological factors and genetic diversity parameters data, a Spearman nonparametric correlation coefficient matrix had been constructed by SPSS. The result showed no significant correlations between the genetic diversity parameters and geoclimatic variables.

## DISCUSSION

In this study, SRAP markers were used to evaluate levels of genetic variation within and among 11 natural populations of Simao pine in China. Previously, three natural populations of Simao pine genetic variation were investigated using nine enzyme systems (*Chen, Zhao & Wang, 2002*). The reported values of the proportion of polymorphic loci, mean genetic distance, and genetic differentiation were all considerably lower than the

results of this study. These differences were likely due to the ability of SRAP markers to detect more loci and higher levels of polymorphism than isozymes markers. One of the benefits of SRAP over other molecular techniques was its sensitivity in stock identification without any upfront knowledge of the species genome, which provided a large number of independent markers that can be rapidly analyzed (*Uzun et al., 2009*; *Soleimani, Talebi & Sayed-Tabatabaei, 2012*; *Ahamd et al., 2014*; *Peng et al., 2015*).

The PIC and polymorphism rate ($P$) were used to measure genetic diversity. A high level of polymorphism was observed in numerous amplification products generated in the course of the analysis. In this study, PIC ranged from 0.43 to 0.77, with an average of 0.63, indicating that the SRAP markers could develop high loci polymorphism useful for genetic variation of accessions studied in this research (*Talebi, Rahimmalek & Norouzi, 2015*). The percentage of polymorphic bands ($P$) of Simao pine was 93.46%, revealing a considerable level of genetic diversity, similar to that of its relatives *Pinus yunnanensis* ($P = 96.43\%$; *Xu et al., 2015*). At a species level, Nei's genetic diversity ($H$) was 0.4567 and Shannon's information indices ($I$) was 0.6484, indicating that Simao pine had a high basis of genetic diversity. Previous studies also showed that narrowly distributed species could have higher genetic diversity (*Dolan et al., 1999*; *Wang et al., 2013*).

In the present study, the total genetic diversity ($H_t$) of the species and genetic diversity within populations ($H_s$) were estimated to be 0.4580 and 0.3801 respectively. Compared to previous SRAP-based studies of pine plants, Simao pine showed higher diversity than *Pinus taxa* ($H_t = 0.2134$; $H_s = 0.3426$; (*Xie et al., 2015*), but lower genetic diversity than khasi pine ($H_t = 0.547$; $H_s = 0.285$; (*Rai, Ginwal & Saha, 2017*). Genetic differentiation ($G_{st}$) was the ratio between the additive genetic variance among populations and the total additive genetic variance (*Wright, 1965*). The study revealed low genetic differentiation ($G_{st} = 0.1701$) among populations of Simao pine. This value was higher than the mean value ($G_{st} = 0.073$) recorded for 121 woody species examined using allozyme markers (*Hamrick, Godt & Sherman-Broyles, 1992*). The low gene differentiation among populations was mainly caused by the high level of gene flow, gene flow could give play to its homogenizing function to prevent differentiation among each population caused by genetic drift. Similar results have been reported for several woody trees that have high levels of genetic diversity, and even in trees with low or moderate levels of genetic differentiation in populations that were located several kilometers away (*Lacerda et al., 2001*; *Arias et al., 2012*). The AMOVA analysis revealed that 86% of the variation was within populations, while the remaining 14% was among populations. This was in agreement with a number of studies where it has been observed that conifers showed high levels of genetic variation within populations and relatively little differentiation among populations (*Hiebert & Hamrick, 1983*; *Agundez et al., 1997*; *Xie et al., 2015*).

Gene flow is the transfer of alleles from one population to another and the study of this is critical for understanding the evolution of plants and population process within and among species (*Grant, 1991*; *Gerber et al., 2014*). Gene flow among populations is mainly produced by foreign genes brought by pollens and seeds for spermatophytes (*Hamrick, 1987*). The $N_m$ of Simao pine was 2.4403, showed that gene exchange among populations was moderately frequent, and weakens genetic differentiation among populations. Gene exchange in the

present study was enough to rule out the possibility that some differentiation among the populations of Simao pine could be due to isolation. This result was consistent with the lack of correlation between genetic and geographic distances ($r = 0.241$, $p = 0.090$), which suggested no significant geographic restriction to gene flow among the populations. The reason for slightly higher $N_m$ may be due to the morphological character of Simao pine, as tall plants benefit from reduced resistance to pollen movement in the air. A long-distance pollen flow also enhanced the gene recombination (*Feng et al., 2009*). Furthermore, a lack of effective geographic isolation may also have contributed to the improvement of gene homogenization among provenances. Simao pine forests were mainly distributed in low mountainous upland in southern China, with a closer distance between provenances.

The genetic relationship among the 11 analyzed populations of Simao pine was represented in a dendrogram showing a clear genetic differentiation of the SM population, which was genetically distinct and could be considered to be more genetically diverse compared to the other ten populations. Principal coordinate analysis (PCoA) was performed to provide spatial representation of the relative genetic distances among populations defined by cluster analysis. The results of UPGMA and PCoA also showed that the assignment of populations was not completely consistent with their geographic origin, a result that has been observed in previous studies (*Liu et al., 2013*; *Cheng, Zheng & Sun, 2015*). This finding indicated that geographical distance had no obvious effect on genetic differentiation of Simao pine populations.

A Bayesian cluster analysis performed with STRUCTURE, showed that the most probable number of genetic groups in the data was two ($K = 2$) for Simao pine. Cluster I was composed of the NE, JD, and ZY populations from the northwest of the Simao pine natural range, collected from regions that had higher altitude (mostly above 1,400 m) and latitude. The ML, JH, Mla, and SM populations which belonged to cluster II were collected from southern regions that had lower altitude (mostly below 1,300 m) and latitude. These results possibly reflected an adaptational difference. It implied that geographical and environmental factors together created stronger and more discrete genetic differentiation than isolation by distance alone.

In the study, there were no significant correlations between the ecological factors and genetic diversity parameters according to Spearman nonparametric correlation analysis. It was suggested that the ecological factors had less effect on genetic diversity. This may be due to the limited distribution, and the small geographical and ecological differences of the studied populations. The high level of genetic diversity in Simao pine is mainly caused by the traits of long-life, generation overlap, wind pollination, and wind-seed dispersal, which lay a broad genetic base for the species (*Chen, Zhao & Wang, 2002*).

Retaining genetic diversity was the key to multi-generation improvement of forest trees and fine provenances should have an extensive genetic base and significant genetic gain (*Feng et al., 2009*). Thorough understanding of the extent and patterns of genetic diversity in Simao pine was essential for its conservation and utilization. The results reported here revealed high genetic diversity at the population level and low genetic diversity at the species level in Simao pine. Neither the percentage of polymorphic loci nor diversity index displayed significant difference among the 11 provenances. Therefore, selective breeding

of Simao pine forests can disregard the limitation of geographic location and should directly select individuals with advanced growth traits. The low genetic differentiation among populations implied that conservation efforts should aim to preserve all existing populations of this species. Based on this, the in situ conservation method was proposed as it is of paramount urgency that sufficient natural population numbers and sizes were conserved to prevent a reduction in genetic diversity. Moreover, in order to achieve effective conservation of germplasm resources, efforts were required to carefully plan and construct pollen and gene banks for Simao pine. In the meantime, natural protection areas should be established to conserve and restore the habitat and populations. In this study, population JC displayed relatively high genetic diversity, and should therefore be a priority site for in situ conservation. In addition, it was important to develop a core collection of Simao pine, which would not only mitigated the pressure of excessive exploitation of wild resources, but can also helped in achieving more effective management and utilization of germplasm.

## CONCLUSIONS

The present study is understood to be the first genetic investigation of Simao pine using SRAP markers to understand distribution and genetic variation. The results indicated that SRAP markers can be efficiently used in the study of genetic diversity and genetic variability of Simao pine. The study revealed high levels of genetic diversity and low or moderate levels of genetic differentiation. Slightly frequent gene flow existed within Simao pine populations, weakening genetic differentiation among these populations. Based on these results, conservation strategies for Simao pine have been formulated.

## ACKNOWLEDGEMENTS

We wish to express our gratitude to Zong Dan, PhD from the Faculty of Forestry, Southwest Forestry University for sharing with us her invaluable advice and knowledge regarding the development and interpretation of POPGEN and GenAIEx results.

### Funding
This study was supported by the National Natural Science Foundation of China (Grant No. 31500536). The funders had no role in study design, data collection and analysis, decision to publish, or preparation of the manuscript.

### Grant Disclosures
The following grant information was disclosed by the authors:
National Natural Science Foundation of China: 31500536.

### Competing Interests
The authors declare there are no competing interests.

## Author Contributions

- Dawei Wang conceived and designed the experiments, performed the experiments, analyzed the data, contributed reagents/materials/analysis tools, prepared figures and/or tables, authored or reviewed drafts of the paper, approved the final draft.
- Bingqi Shen performed the experiments, analyzed the data, contributed reagents/materials/analysis tools, prepared figures and/or tables, approved the final draft.
- Hede Gong analyzed the data, authored or reviewed drafts of the paper, approved the final draft.

## Field Study Permissions

The following information was supplied relating to field study approvals (i.e., approving body and any reference numbers):

Field experiments were approved by the Southwest forestry university (project number: 2013Y121).

## Data Availability

The raw data are available in the Supplemental Files.

## Supplemental Information

Supplemental information for this article can be found online at http://dx.doi.org/10.7717/peerj.6529#supplemental-information.

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
