# Peer review of "Genetic diversity of Simao pine in China revealed by SRAP markers"

_PeerJ, doi:10.7717/peerj.6529_

## Round 0.1 · original submission · Minor Revisions

Dear Dr. Dawei Wang

Your MS entitled “Genetic diversity of Simao pine in China revealed by SRAP markers”, has been reviewed by two independent reviewers, which raised several important issues (see reviewers comments). Based on that, I suggest that you include minor revisions. Particularly, I would like to highlight the need to deepen the discussion, regarding the population structure and driving ecological factors. Please follow the recommendations of the three reviewers.

Also, please provide a new version edited by a fluent English speaker.

Looking forward to receive the revised version,

Sincerely

Ana I. Ribeiro-Barros

Reviewer 1 ·

Basic reporting

This manuscript investigated the genetic diversity of Simao pine based on the SRAP markers. The authors found that this important conifer species have the high level of genetic diversity and low level of genetic differentiation among populations. They also proposed the some conservation and management strategies. Actually, the manuscript is interesting for the general reader of Peer J.
There are some questions which need to address before acceptance in the Journal.
Line 32, “the genetic diversity of the entire remaining natural populations…”, what is mean of the entire remaining natural populations? It seems that the means of the sentence is unclear, which should need to revise.
Line 39-40, Results. The 11 populations reveal a relatively low level of genetic diversity in the species. Results show a high level of genetic diversity (PPB = 94.45%, H = 0.4567, I = 0.6484). The means of these two sentences are confused. Please revise it carefully.
Line 42, AMOVA analysis changed to AMOVA.
The cited literatures style, e.g., Line 59 (Wang et al., 2012) and others. The comma should be body not italic, please carefully revise the whole manuscript.

In the whole manuscript, there are some mistakes and errors of the grammars and sentences. I think the authors could thoroughly revise the manuscript in the help of a native English speaker. Some spaces should be noted. E.g., Longitude(° E) Latitude(° N) in Table 1.

In addition, I suggest that the authors could run the STRUCTURE analysis for their datasets and they could discuss the population structure furtherly.

Experimental design

useful.

Validity of the findings

Interesting and useful.

Additional comments

.

Reviewer 2 ·

Basic reporting

Table 2
Please note the direction of the primers' sequences

Figure 2
Please note the four groups mentioned in the manuscript.

Figure 9
Adding a Chinese map will make clear about the locations of sampling in this study.

Experimental design

no comment

Validity of the findings

no comment

Additional comments

1 Authors may discuss about how ecological factors influence the level of genetic diversity of Simao pine populations in nature.
2 Please give some possible reasons why Gst of Simao pine is different from some species or similar with some species.

Reviewer 3 ·

Basic reporting

In general, the English needs to be improved. There are many places with items such as missing punctuation and use of the term "use". These are not major problems, but need attention to improve the readability of the manuscript. Please see other specific comments on the attached manuscript.

Experimental design

no comment

Validity of the findings

no comment

Additional comments

Thank you for your contribution. In general,the manuscript is clearly written in professional, unambiguous language. If there is a weakness, it is in the missing punctuation and use of the term "use". (as I have noted above) which should be improved upon before Accepted. What environment were these Simao pine populations? Full sun or shade? Did this affected genetic clustering? And I think some original electrophoresis map is needed to public.

Annotated reviews are not available for download in order to protect the identity of reviewers who chose to remain anonymous.

---

## Round 0.2 · Minor Revisions

Dear Dr Wang,

Thank you for sending the revised version of your MS which is now acceptable for publication. Nevertheless, before accepting it I would like to ask you to edit the text in order to correct the few typos and change the verbs from present to past tense.

After these changes are included I will accept immediately the paper.
Sincerely

Ana Ribeiro-Barros

Reviewer 1 ·

Basic reporting

Clearly.

Experimental design

Reasonable.

Validity of the findings

accurate and interesting.

Additional comments

I have checked the manuscript carefully again and found it thoroughly revised the questions that I have suggested previously. So I recommend accept the manuscript after a minor question addressed: L 120, please authors add some parameters of STRUCTURE analysis for the datasets.

---

## Round 0.3 · Minor Revisions

Dear Dr Wang

Thank you for sending the revised version of your MS. At this stage, the MS still needs a deep language editing, particularly in the use of past tense (instead of present tense) in the abstract, methods, results and discussion. Please address carefully this concern.

Best regards

Ana I. Ribeiro-Barros

---

## Round 0.4 · Minor Revisions

Dear Dr Dawei Wang

The revised version of the MS has improved considerably, but is not yet acceptable for publication due to language editing problems, for example:

- "Introduction" is writen "Itrduction"

- Last paragraph of the introduction is still in the present tense form: "In the present study, SRAP markers are used to ….""The study aims to evaluate …." etc

If you agree, I will give you the last chance to solve the language editing problem.

Best regards
Ana Ribeiro-Barros

---

## Round 0.5 · accepted · Accept

Dear Dr. Dawei Wang

It is my pleasure to inform you that your MS is not accepted to be published in PeerJ.

Thank you for considering this journal to publish your work,

Sincerely
Ana Ribeiro-Barros

#